# Cetaceans Humerus Radiodensity by CT: A Useful Technique Differentiating between Species, Ecophysiology, and Age

**DOI:** 10.3390/ani12141793

**Published:** 2022-07-13

**Authors:** Francesco Maria Achille Consoli, Yara Bernaldo de Quirós, Manuel Arbelo, Stefania Fulle, Marco Marchisio, Mario Encinoso, Antonio Fernandez, Miguel A. Rivero

**Affiliations:** 1Veterinary Histology and Pathology, Atlantic Center for Cetacean Research, Institute of Animal Health and Food Safety (IUSA), University of Las Palmas de Gran Canaria (ULPGC), 35400 Las Palmas, Spain; francesco.consoli@unich.it (F.M.A.C.); manuel.arbelo@ulpgc.es (M.A.); antonio.fernandez@ulpgc.es (A.F.); miguel.rivero@ulpgc.es (M.A.R.); 2Department of Neuroscience Imaging and Clinical Sciences, University G. D’Annunzio, 66100 Chieti, Italy; stefania.fulle@unich.it; 3Department of Integrative Physiology, University of Colorado Boulder, Boulder, CO 80303, USA; 4Department of Medicine and Aging Sciences, Center for Advanced Studies and Technology (CAST), University G. D’Annunzio, 66100 Chieti, Italy; marco.marchisio@unich.it; 5Hospital Clínico Veterinario, Facultad de Veterinaria, Universidad de Las Palmas de Gran Canaria, 35413 Las Palmas, Spain; mencinoso@gmail.com

**Keywords:** bone, computed tomography, radiodensity, cetaceans

## Abstract

**Simple Summary:**

Cetaceans are marine mammals whose tissues have undergone an evolutionary process to adapt to life in the aquatic medium. In addition, depending on the species, the adaptations will have developed differently. However, some features of the bones of these animals are still poorly studied, and a good technique for analysing this tissue is computed tomography. With this, it is possible to measure bone density, compare bone density between distinct species, and then relate it with the ecophysiological aspects of the animal. For this reason, this study aimed to create a protocol to measure the bone radiodensity of the humerus of cetaceans through computer tomography. The purpose was to be able to make comparisons between different species to advance the understanding of this cetacean tissue. This project considered two species with different characteristics and behaviours: the Atlantic spotted dolphin (*Stenella frontalis*) and the pygmy sperm whale (*Kogia breviceps*). The main results observed were significant differences in bone density between different types of bone (trabecular and cortical bone) and various portions of the humerus within a given species. Such differences were also observed between species with varying habits of diving (shallow divers and deep divers) and distinct swimming habits (swimming style with low-medium energy expenditure and swimming style that requires medium-high energy consumption rates). These results allow us to consider the unique adaptations of this tissue to adapt to varying lifestyles of these animals in the aquatic ecosystem. Therefore, this technique will allow for comparisons to distinguish different characteristics related to species, age, diving behaviour, and swimming style.

**Abstract:**

Cetaceans are mammals that underwent a series of evolutionary adaptations to live in the aquatic environment, including morphological modifications of various anatomical structures of the skeleton and their bone mineral density (BMD); there are few studies on the latter. BMD is related to the radiodensity measured through computed tomography (CT) in Hounsfield units (HU). This work aimed to test and validate the usefulness of studying humeral bone radiodensity by CT of two cetacean species (the Atlantic spotted dolphin and the pygmy sperm whale) with different swimming and diving habits. The radiodensity was analysed at certain levels following a new protocol based on a review of previous studies. Humeral radiodensity values were related to four aspects: species, diving behaviour, swimming activity level, and age. We observed that the consistent differences in the radiodensity of the cortical bone of the distal epiphysis between animals of different life-history categories suggest that this bone portion could be particularly useful for future ontogenetic studies. Hence, this technique may be helpful in studying and comparing species with different ecophysiologies, particularly distinguishing between swimming and diving habits.

## 1. Introduction

Cetaceans are mammals that underwent a series of evolutionary adaptations to live in the aquatic environment [1], including morphological modifications of various anatomical structures of the skeleton and their bone density [2,3,4,5,6,7,8,9,10,11]. However, there are few studies on the latter.

Bone density, or bone mineral density (BMD), is the mass of minerals per volume of bone [12,13] and is related to the radiodensity measured through computed tomography (CT) in Hounsfield units (HU) [14]. HU indicates the tissue’s ability to attenuate X-rays that pass through it and depends on the tissue’s nature (e.g., type of bone and bone compartment element) [15,16,17] and density [12,18,19,20,21,22,23]. Differences in this measure also exist between species [15,24,25,26,27,28], sex [29,30,31], and age [32,33,34], as well as mechanical stress on the bones during movements [28,35,36] or in the face of various pathologies [37,38,39]. Therefore, investigations on bone density could help understand the evolutionary phenomena and adaptations related to lifestyle and ontogenesis in different animal species and contribute to diagnosing pathologies.

The humerus, in terrestrial mammals, is a long bone whose medullar cavity has bone marrow [40]. In cetaceans, the humerus is a relatively short bone that lacks the medullar cavity. However, its bone compartments, cortical and trabecular, can be easily distinguished [3]. Cetaceans’ humerus joints distally with the radius and ulna and proximally with the scapula through the glenohumeral joint. The latter is not as mobile as in terrestrial mammals [40], but it still articulates, allowing some range of fin movement during swimming [3]. This feature is essential to evaluate how physiological joint movements can cause mechanical stress with consequences on bone radiodensity as in terrestrial mammals and humans [28,35,36,41,42], and how some joint pathologies may affect bone radiodensity similarly to other species [37,43,44,45,46,47].

Some previous literature investigating the cetacean pectoral fin bones analysed the samples using radiological techniques focusing on sample variation with animal growth, the degree of ossification, and radiodensity measured by a dual-energy X-ray absorptiometry (DXA) device. These studies described changes in radiodensity related to growth [5,9,48,49] and sex [6,7]. Any apparent differences in bone radiodensity were found between both sexes in beluga whales, long-finned pilot whales, and fin whales [6,7].

The infraorder Cetacea includes many species with different behaviours in diving and swimming. For example, the Atlantic spotted dolphin (*Stenella frontalis*) of the Delphinidae family is considered a shallow diver that usually swims to a depth of 20 m, although it can reach a depth of 60 m [50,51,52], and its swimming style requires medium-high energy consumption rates [53,54,55]. On the other hand, the pygmy sperm whale (*Kogia breviceps*) of the Kogiidae family is considered a deep diver that usually dives to depths of 500–1000 m [56,57,58,59] and its swimming style entails low-medium energy expenditure [53,60,61]. Such differences between these two cetacean species might be reflected in their bone radiodensities. 

It could be helpful for future studies to establish standardised techniques that allow for the evaluation of how bone density is related to different diving habits and swimming styles and to evaluate the degree of growth of an animal and the bone development related to age [62]. The latter would provide data critical for population studies. Furthermore, it is important to create references for radiodensity data for future comparative pathological studies in these species, such as for dysbaric osteonecrosis [63].

As such, this work aims to validate and obtain preliminary results for future studies, in that two cetacean species (the Atlantic spotted dolphin and the pygmy sperm whale), with different swimming and diving habits, were chosen as specimens to execute this aim.

## 2. Materials and Methods

### 2.1. Animals and CT Scan

The animals included in this study were 15 Atlantic spotted dolphins of 55–185 cm in length (two calves, four juveniles, and three adults) and seven pygmy sperm whales 158–343 cm in length (four calves and three adults) (Table 1), stranded dead in the Canary Islands between 2018 and 2020. The required permission for the management of stranded cetaceans was issued by the environmental department of the Canary Islands Government and the Spanish Ministry of Environment. No experiments were performed on live animals for this study.

At necropsy, a preservation code between 1 and 5 was assigned to each carcass (1 for very fresh corpses and 5 for animals in an advanced state of decomposition) following the international standardised cetaceans post-mortem studies protocol [64]. The various specimens were divided by life history categories: (1) calves, animals that presented with milk in the stomach or were of the approximate size of a suckling calf for that species; (2) juveniles, sexually immature animals with a length higher than a calf but shorter than an adult; (3) adults, sexually mature animals. For the Atlantic spotted dolphin, length references for each age category were taken from a study of 89 Atlantic spotted dolphins in the Canary Islands [65]. Unfortunately, for the pygmy sperm whale, there are no similar works with the population of the Canary Islands. However, four animals were described as calves as they were shorter than 2 m long, comparable to a calf’s size described elsewhere in the literature [56,66].

The right or left thoracic limb (i.e., flipper), usually the most intact one and of most accessible access as both limbs have similar radiodensities [6,7], was dissected from all animals by removing the *rhomboideus, serratus ventralis, serratus dorsalis, pectoralis brachiocephalicus, coraco brachialis*, and *latissimus dorsi muscles*. The neurovascular structures were removed close to the bone surface without damaging the bone. All dissected fins were examined macroscopically during the necropsy and by CT scan. Humeri with bone lesions were discarded from this study. Flippers were scanned by a 16-slice helical CT scanner (Toshiba Astelion, Toshiba Medical System, Madrid, Spain) at the didactical veterinary hospital HCV (Hospital Clínico Veterinario) of the ULPGC (Universidad de Las Palmas de Gran Canaria) (Table 2). Single flippers were positioned on the CT table with the humeral head facing anteriorly, the ultimate distal phalanxes oriented posteriorly, and the cranial edge of the flipper lying as if the entire animal were in left lateral decubitus. A bone filter (Convolution Kernel FC30, Toshiba Astelion, Madrid, Spain) was used, and then the row data files were converted to DICOM files and analysed by Osirix MD © software (Pixmeo company, Geneva, Switzerland).

The humeral length was measured on images obtained with CT along the longitudinal axis of the bone, taking the proximal apex of the tubercle and the distal apex of the trochlea as the two opposite humerus endpoints [67,68]. The radiodensity was analysed at certain levels following a new protocol that was based on a review of previous studies found in the literature [15,20,22,23,27,28,31,32,41,44,68,69,70,71,72,73,74,75,76,77].

### 2.2. Bone Radiodensity Protocol

The bone radiodensity was measured at four different portions (Figure 1A):(1)Humeral distal epiphysis: this bone portion runs between the transverse plane passing by the distal apex of the bone and the transverse plane passing by the apex of the ulna olecranon tuberosity.(2)Humeral body: this bone portion runs between the transverse plane passing by the apex of the ulna olecranon prominence and the transverse plane passing by the humeral neck.(3)Humeral head: this bone portion runs from the transverse plane, passing by the humeral neck, the groove between the humeral head and the humeral tubercle, and the proximal end of the articular surface of the humeral head.(4)Humeral proximal epiphysis: this bone portion runs from the transverse plane passing by the humeral neck to the apex of the humeral tubercle. Not including the humeral head.

From each portion, three equidistant sections were obtained. First, the cortical and trabecular bones were visually divided by comparing the more radiopaque bone areas, i.e., the cortical bone, from the more radiolucent ones, i.e., the trabecular bone. (Figure 1B, Table 3 and Table 4). Next, the humeral head (3rd portion) was divided into two sub-portions, the lateral and medial (Figure 1C, Table 3 and Table 4). This decision was made according to the observation that some land mammals showed differences in bone density on the articular surfaces [27,78]. Then, each sub-portion was subdivided into three sections similar to the previous humeral portions. Each section was parallel to the division plane that separated the head from the rest of the humerus.

### 2.3. Data Analysis 

The radiodensity average and standard deviation in HU were calculated for the different sections of each bone portion. Linear regression studies between the average radiodensity for the other bone sections and humeral length were calculated using Excel © 2013 (15.0.5172.1000). The level of statistical significance was set at 0.05.

## 3. Results

Humeral length measurements showed smaller sizes in calves and juveniles compared to adults. This result is because the humeral longitude increases as the animal grow from calf to adult when the size begins to stabilise.

Linear regressions for each portion of the humerus, type of bone, and section were calculated against the humeral length (Table 3). Below, we focus on the results more relevant for this study. Animals with conservation code 5 presented very different HU values than their counterparts and were excluded from the study.

The radiodensity values (average ± standard deviation) were also calculated for each life history category (Table 4).

### 3.1. Atlantic Spotted Dolphins

#### 3.1.1. Distal Epiphysis

The cortical bone radiodensity presented a strong positive relationship with humeral length (*p* < 0.001), while the first and second trabecular bone sections did not. As a result, the radiodensities of the cortical and trabecular bone were similar in short humeri (cortical bone 740.07 ± 90.31 HU and trabecular bone 689.43 ± 47.45 HU), but the cortical bone became denser with humeral length (905.59 ± 21.41 HU) (Table 3) (Figure 2A).

#### 3.1.2. Humeral Body

The humeral body presented the most relevant radiodensity differences between the cortical and trabecular bone, with the cortical bone consistently denser than the trabecular bone. The radiodensity did not change significantly with humeral length (Table 3) (Figure 2B).

#### 3.1.3. The Medial Aspect of the Humeral Head

No evident differences in radiodensity were observed between the medial aspect of the humeral head cortical and trabecular bones (Figure 2C). Both become significantly denser with humeral length (Table 3). The shorter humeri presented a 478.50 ± 232.27 HU radiodensity, while the longer humeral radiodensity was 657.36 ± 111.97 HU (Figure 2C).

#### 3.1.4. The Lateral Aspect of the Humeral Head

The radiodensity of cortical and trabecular bone increased significantly with humeral length in the lateral portion of the humeral head (Table 3) (Figure 2D). However, the cortical bone sections presented a steeper slope. The shortest humeri had very similar cortical and trabecular radiodensity values with an average of 560.09 ± 308.25 HU, while the third section of the cortical bone of the longer humeri was 1048.69 ± 183 HU (Figure 2D).

#### 3.1.5. Proximal Epiphysis

The cortical and trabecular bone presented similar radiodensity values in the proximal epiphysis with no significant changes with humeral length, except for the second section of the trabecular bone, which became less radiodense with humeral length (R^2^ = 0.46, *p* = 0.037) (Table 3).

### 3.2. Pygmy Sperm Whale

#### 3.2.1. Distal Epiphysis

The cortical bone was denser than the trabecular bone at the distal epiphysis. No significant differences in radiodensity with humeral length were observed except for the third section of the cortical bone, which became more radiodense with humeral length (R2 = 0.72 and *p* = 0.019) (Table 3) (Figure 3A).

#### 3.2.2. Humeral Body

The humeral body presented the most relevant radiodensity differences between the cortical and trabecular bone, with the cortical bone consistently denser than the trabecular bone. Furthermore, this difference became larger with humeral length, as the cortical bone increased in radiodensity while the trabecular bone decreased with humeral length. However, these trends were not statistically significant (Table 3) (Figure 3B).

#### 3.2.3. The Medial and Lateral Aspects of the Humeral Head

The radiodensities of the cortical and trabecular bone were similar at the medial aspect of the humeral head with no statistically significant regressions with humeral length. Similar results were observed in the lateral aspect of the humeral head (Table 3) (Figure 3C,D).

#### 3.2.4. Proximal Epiphysis

The results obtained for the radiodensity of the proximal epiphysis sections did not show linear regressions worthy of description and did not show a similar pattern between them (Table 3).

### 3.3. Comparison between Species

The results of the same sections in the two different species were compared (Figure 4). The radiodensities of the other portions of the humerus were consistently higher in the Atlantic spotted dolphin than in the pygmy sperm whale. The largest differences were observed at the humeral distal epiphysis and body (Figure 4A,B).

## 4. Discussion

This study describes and validates a simple and effective technique to study the bone radiodensity of the humerus of cetaceans through CT. Observations made in this study provide preliminary results for future investigations. The technique used in this work enabled bone radiodensity evaluation at different levels of the humerus, allowing for the identification of the specific region of the bone to be studied specific to a given research question. Hence, studying radiodensity is helpful for different purposes, such as studying the density of various regions of bone tissue, as well as changes in bone density between and within regions of bone as a result of various eco-physiological stimuli.

One of the innovations of this protocol is dividing the humerus into four portions delineated by anatomical points as reference limits [69,70,71,74]. Hence, we studied the entire humerus rather than some areas of the humerus, as performed in previous studies of terrestrial mammals [15,20,22,23,27,28,31,32,41,44,68,69,70,71,72,73,74,75,76,77]. Additionally, each portion of the humerus was subdivided into three equidistant sections to evaluate whether noticeable changes exist in bone radiodensity within the same portion of the bone [22,31,68,73,76]. We also analysed the cortical bone and trabecular bone independently as these two tissues react differently to ageing and mechanical stress [15,23,70,75,76]. We deemed such a detailed approach important as it has been previously reported that different factors affecting the density of the humeral bone may impact various portions of the bone [15,17,20,24,25,29,30,32,33,35] differently.

To our knowledge, none of the previous studies offered such a comprehensive analysis of mammalian bones. Moreover, very few papers reported analyses on bone density and the ossification of the bones of the thoracic limb, specifically in cetaceans [5,6,7,9,48,49,62]. Indeed, most of these studies only considered the total density of the whole bone. It has to be acknowledged that Felt et al. (1965) [6] did measure the bone radiodensity at the humeral body region, perpendicular to the longitudinal axis, but without considering possible variations along the humerus. In addition, Felt et al. (1965) [6] used a relatively simple technique at that time, physically sectioning the bone, causing artefacts, before analysing the specimens by dual-energy X-ray absorptiometry. Rather, our protocol kept a given bone intact and divided the sections virtually by using computer software programs. This method enables the study of thinner sections, prevents possible artefacts, and offers much more precise results than Felt et al. (1965) [6].

The analyses on the humerus led to different radiodensity values on each portion, as in terrestrial mammals [15,22,23,31,68,69,70,72,73,76]. Furthermore, analyses performed on individuals of different ages of the same species and individuals of different species led to the observation of significant differences in bone density in one portion of a given bone or another. Furthermore, the technique was validated by the consistency of the results obtained in specimens presenting the same characteristics (age, species, and behaviour) (Table 4). It is important to remember that bone density is affected by age as well as mechanical stress brought on by differing diving or swimming habits [15,20,27,28,31,32,70,76].

One of the most evident results of this manuscript’s technique is that the humeral radiodensity differed between species. The shallow-diving Atlantic spotted dolphin had a higher radiodensity than the deep-diving pygmy sperm whale, in agreement with the literature reporting bone density values for shallow and deep divers [1,6,7,8,10]. This aspect demonstrates the accuracy and potential usefulness of this methodology in studying species with different ecophysiological adaptations.

The radiodensity difference between the two species was higher in the body of the humerus (Figure 4). This observation agrees with the demonstration that aquatic species have undergone a readaptation of the morphology of the bone tissue with consequent weight loss during evolution [8]. Furthermore, deep-diving species have suffered a loss of bone density through evolution, likely caused by a physiological process that can be described as osteoporosis [4,6,7,8]. Furthermore, the radiodensity of the trabecular bone at the distal epiphysis decreased further with humeral length. This trend was not observed in the Atlantic dolphin, a shallow diving species, suggesting that the pigmy sperm whale loses radiodensity as it develops its deep-diving abilities with age. Therefore, considering these findings, future studies should focus on measuring humeral radiodensity to assess potential differences between species.

Differences between these two species were also noticed in the lateral aspect of the humeral head. Based on the observation of many scapulohumeral joints, our understanding is that the lateral portion of the humeral head is the aspect of the humerus most in contact with the glenoid cavity of the scapula [3]. Thus, this portion should be the most exposed to mechanical stress. According to the literature, relative to terrestrial mammals, bone radiodensity is not uniformly distributed along the joint surface. It depends on the type of movement the bones are subjected to and the intensity of physical exercise [28,35,36,41,42,78,79,80,81]. The Atlantic spotted dolphins have a high-energy swimming behaviour [53,55], with fast changes in directional swimming where the pectoral flippers play a steering role [61,82,83,84,85,86,87,88] in contraposition to the pigmy sperm whale, which has a low-energy swimming behaviour [53,54,55] with less abrupt directionality changes in their swimming behaviour [61,82,83,84,85,86,87,88]. As a result of this difference in swimming behaviour, the cortical radiodensity of the lateral humeral head was increased (as compared to the humeral medial aspect) in the Atlantic spotted dolphin but not in the pygmy sperm whale.

Bone radiodensity changed for most aspects and sections with humeral length. As bones grow in length as the animals grow [40,89], the humeral length can be an indirect indicator of age. Indeed, calves presented the shortest humerus while adults presented the longest. The cortical bone of the three sections of the distal epiphysis of the Atlantic spotted dolphin gave the highest positive linear regression coefficient with humeral length, suggesting that the radiodensity of this region might be the most important in determining age differences. If this is proven true, radiodensity could be used in future studies to determine age or age categories more simply and less expensively than tooth analyses or genetic techniques [90,91]. However, differences in accuracy between the techniques may likely exist and are yet to be determined. We did not observe in pigmy sperm whale specimens the same strong correlation between the radiodensity of the cortical bone at the distal epiphysis and the humeral length as we did in Atlantic spotted dolphin specimens. These could be due to the low sample size and a higher number of calves within the animals observed. Future studies with more juvenile and adult pygmy sperm whales are needed to confirm if the distal epiphysis is the best region to study bone ontogeny in this species.

The different portions of the humerus described in this protocol do not follow the anatomical portions established by the NAV (Nomina Anatomica Veterinaria) nomenclature [92], as the metaphysis was not considered as separation limits for the portions. The metaphysis tends to ossify after a certain age [40] and becomes impossible to identify through computed tomography. Other techniques, such as histological techniques, could perhaps help study bone and its density. Histological techniques also have limitations, such as the inability to study the entire humerus and the potential changes in the different regions. Future studies could benefit from combining both techniques, first using our CT scan protocol for an overview of the bone and then locating abnormalities or locations of interest for histological sampling.

This protocol could be used with all cetacean species, regardless of the humeral anatomical morphology and age. This possibility is advantageous due to the difficulty of obtaining animals of every life history category and species and in a good conservation code (1 or 2) as the studies are performed on stranded animals. One limitation of our methodology is the study of radiodensity from animals in an advanced decomposition (conservation code 5). These humeri presented very different values compared to the other animals, especially in the proximal epiphysis, suggesting that advanced decomposition can affect bone density in agreement with the literature [93,94,95,96,97].

In future studies, this methodology will be advantageous if combined with other techniques such as histology and validated if extended to a larger number of samples. The analyses of radiodensity on various individuals could help to understand the different characteristics that influence radiodensity, such as the individuals’ age, species, and species-specific habits. Future studies with a higher sample size could help determine physiological ranges of bone radiodensity related to the portion of the bone, the species, and the degree of animal growth, contributing to the diagnosis of bone pathologies in these animals. Those studies should also verify if these values are valid for diagnostic purposes in animals with a decomposition code of 4 or 5.

## 5. Conclusions

In summary, with this methodology, it was possible to describe different humeral radiodensity values as related to diving behaviour, with deep-diving species having lower radiodensity values than shallower-diving species. The variations in radiodensity in the lateral aspect of the humeral head indicated a more active swimming style, with the Atlantic spotted dolphin presenting with a higher number of variations than a pygmy sperm whale. Humeral radiodensity changed with age, especially in the cortical bone of the distal epiphysis, suggesting that this bone aspect should be selected to be analysed in future ontogenic studies. As such, this technique holds the potential to study and compare species with different ecophysiological niches, particularly diving and swimming habits.

## Figures and Tables

**Figure 1 animals-12-01793-f001:**
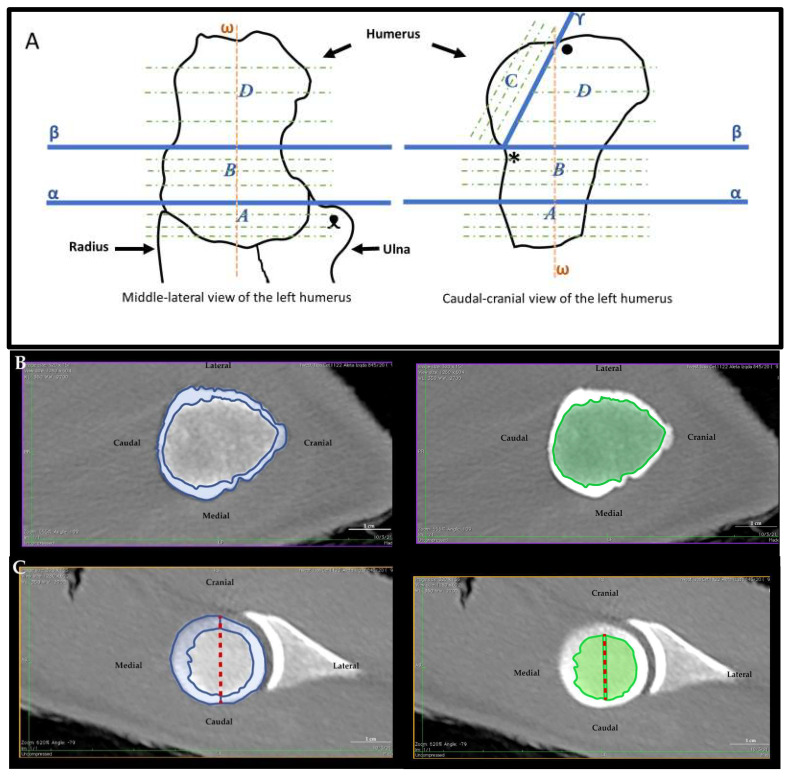
Illustration of the applied protocol. (**A**): Schematic representation of the applied protocol to obtain the four portions of the humerus: it shows the longitudinal axis of the humerus (ω), the plane (β) passing through the neck of the humerus (*), and the plane (α) passing tangentially to the olecranon of the ulna (ᴥ). Proximally, the plane (Υ) passes through the neck of the humerus (*) and the notch (•) between the head of the humerus and the tubercle. These planes delimit the portions: (*A*) distal epiphysis; (*B*) body; (*C*) head of the humerus; (*D*) proximal epiphysis consisting entirely of the tubercle. The green dashed lines represent the various sagittal planes equidistant in the different portions of the various measurements. (**B**): Example of a humeral section (third transversal section of the humeral body of Atlantic spotted dolphin). The area corresponding to the cortical bone can be seen highlighted in blue, and the area corresponding to the trabecular bone in green. (**C**): Example of a humeral head section (second transversal section of the humeral head of Atlantic spotted dolphin). The section was divided into lateral and medial aspects following the midline (red dashed line). In blue are the two areas of the cortical bone (lateral and medial). In green are the two areas of the trabecular bone (lateral and medial).

**Figure 2 animals-12-01793-f002:**
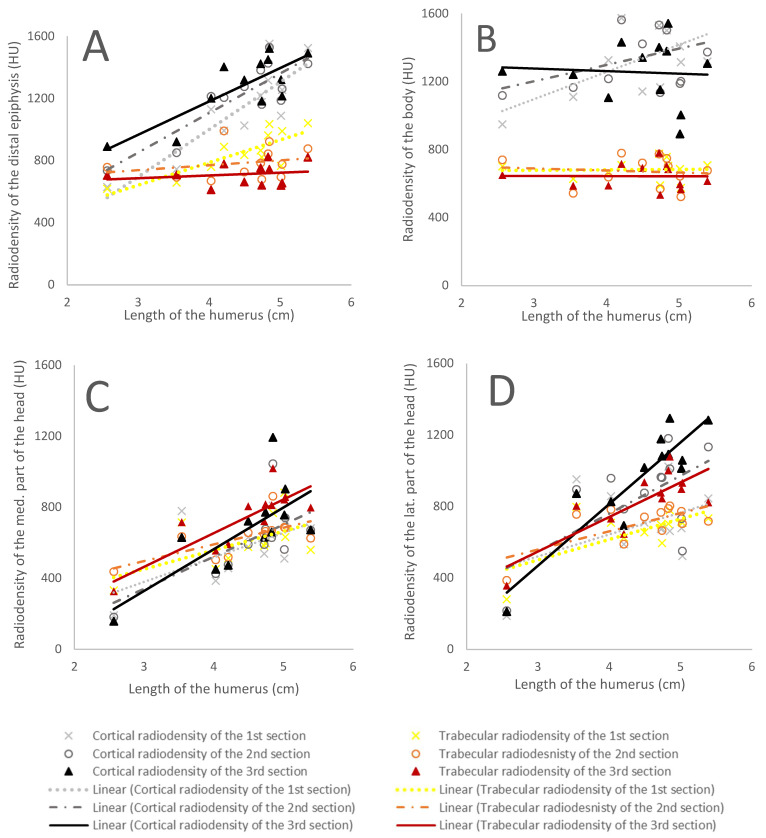
The graphs show how the different humeral portion radiodensities changed in relationship with the humeral length in the Atlantic spotted dolphins: (**A**) the humeral distal epiphysis, (**B**) the humeral body, (**C**) the medial aspect of the humeral head, and (**D**) the lateral aspect of the humeral head. Black and grey lines represent cortical bone density values (CD), while the red, orange, and yellow lines represent trabecular bone density values (TD). Dotted lines represent the regression lines for the first section, dashed lines are for the second section, and the continuous line depicts the third section.

**Figure 3 animals-12-01793-f003:**
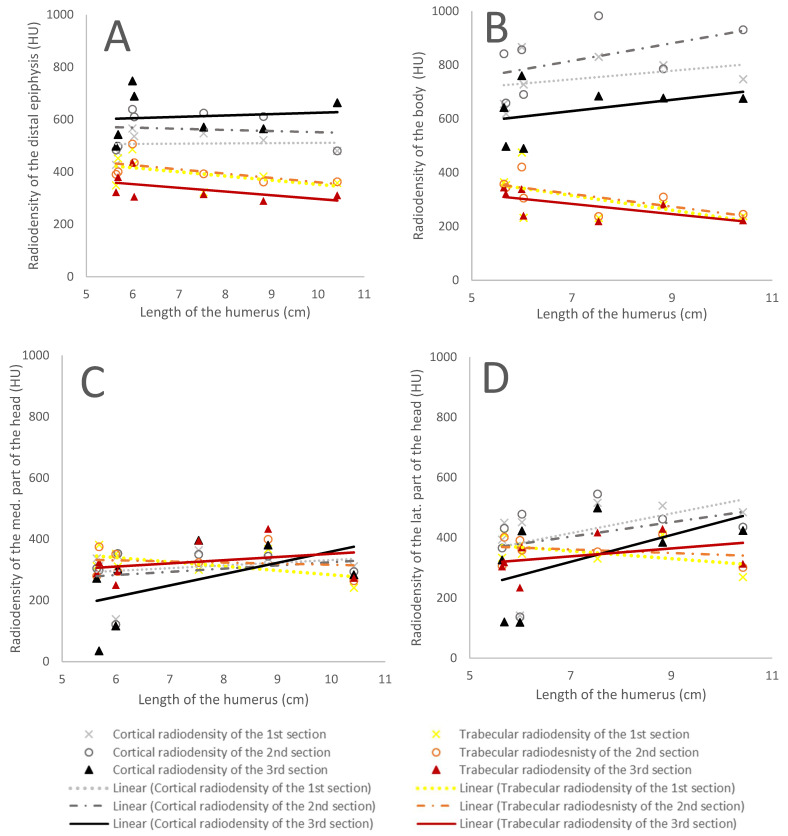
The graphs show differences in radiodensity with humeral length in the pigmy sperm whales: (**A**) the humeral distal epiphysis, (**B**) the humeral body, (**C**) the medial aspect of the humeral head, and (**D**) the lateral aspect of the humeral head. Black and grey lines represent cortical bone density values (CD), while the red, orange, and yellow lines represent trabecular bone density values (TD). Dotted lines represent the regression lines for the first section, dashed lines are for the second section, and the continuous line depicts the third section.

**Figure 4 animals-12-01793-f004:**
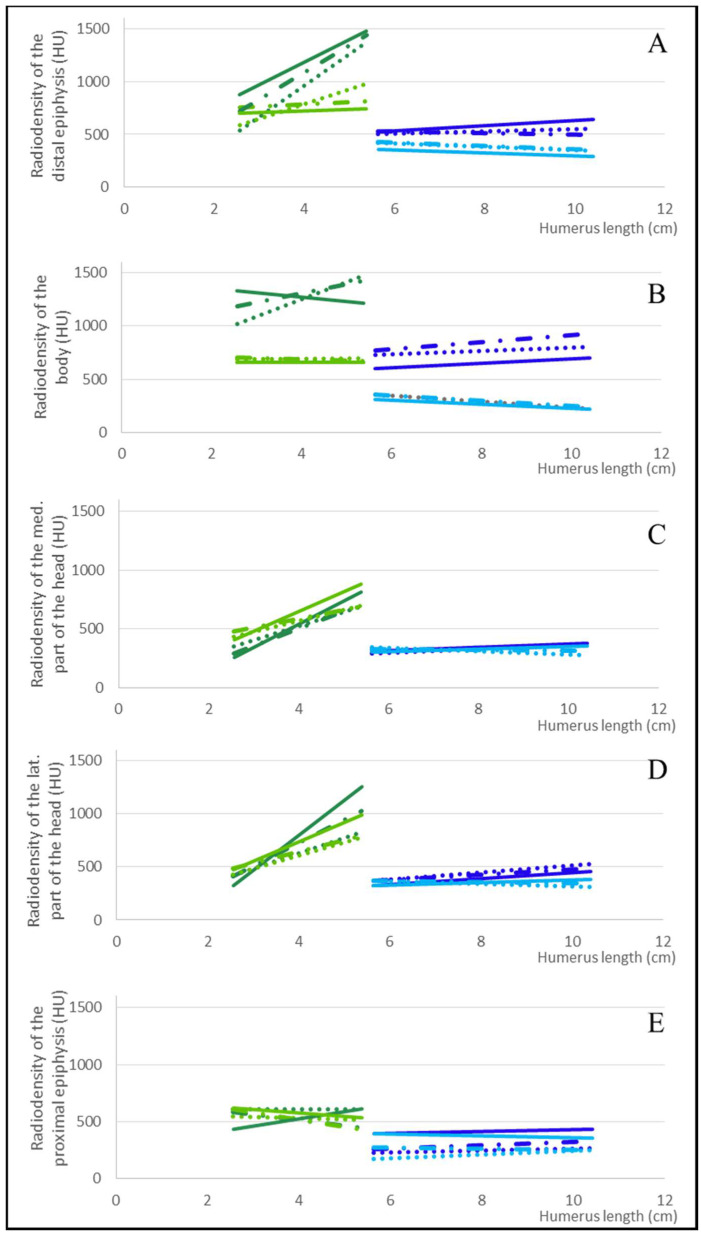
The graphs compare the radiodensity regression lines between the Atlantic spotted dolphin and the pygmy sperm whale at the (**A**) distal epiphysis of the humerus, (**B**) humeral body, (**C**) medial aspect of the humeral head, (**D**) lateral aspect of the humeral head, and (**E**) proximal epiphysis. The regression lines of the cortical bone of the Atlantic spotted dolphin are dark green in colour, and the regression lines of the trabecular one of the same species are light green. On the other hand, the regression lines of the cortical bone of the pygmy sperm whale are dark blue, and the regression lines of the trabecular bone of the same species are light blue.

**Table 1 animals-12-01793-t001:** Animals from which the flippers were dissected and analysed. NV: Not Valuable means these values could not be measured, because of logistic problems or because the animal corpse was in poor conservation condition. C.C.: Conservation Code.

Code	Species	Sex	Life History Category	Animal Length (cm)	Weight (kg)	C.C.	Humerus’ Length (cm)
CET 957	*Kogia breviceps*	Male	Calf	165	67.5	3	5.64
CET 887	*Kogia breviceps*	Female	Calf	164	56.1	2	5.68
CET 704	*Kogia breviceps*	Female	Calf	158	72	2	5.99
CET 883	*Kogia breviceps*	Male	Calf	175	72	4	6.03
CET 916	*Kogia breviceps*	Male	Adult	240	NV	3	7.53
CET 736	*Kogia breviceps*	Male	Adult	261	134	3	8.82
CET 764	*Kogia breviceps*	Male	Adult	343	NV	4	10.41
CET 918	*Stenella frontalis*	Female	Calf	55.8	NV	3	2.56
CET 912	*Stenella frontalis*	Male	Calf	82.6	8	5	2.74
CET 951	*Stenella frontalis*	Female	Calf	122	NV	2	3.54
CET 882	*Stenella frontalis*	Male	Calf	136.5	NV	5	4.15
CET 884	*Stenella frontalis*	Female	Calf	145	36.5	2	4.49
CET 854	*Stenella frontalis*	Female	Juvenile	128	NV	3	4.20
CET 835	*Stenella frontalis*	Male	Juvenile	165	50.4	2	5.01
CET 945	*Stenella frontalis*	Male	Adult	177.5	80	4	4.02
CET 1026	*Stenella frontalis*	Male	Adult	116	NV	5	4.93
CET 872	*Stenella frontalis*	Female	Adult	168	50	3	4.72
CET 917	*Stenella frontalis*	Female	Adult	179	NV	4	4.74
CET 885	*Stenella frontalis*	Male	Adult	185	73.5	3	4.82
CET 871	*Stenella frontalis*	Female	Adult	176	35.6	4	4.84
CET 834	*Stenella frontalis*	Female	Adult	175	65.4	2	5.02
CET 847	*Stenella frontalis*	Male	Adult	179	NV	2	5.38

**Table 2 animals-12-01793-t002:** The table shows the values established in the protocol for CT scan flippers.

Manufacture	Toshiba
Model Name	Astelion 16
Scan Option	HELICAL CT
Slice Thickness	1 mm
Slice increment	0.80 mm
KvP (Kilo Volt Peak)	120 Kv
Exposure	75 mAs
Filter Type	Large
Convolution Kernel	FC30
Patien position	HFDL (Head Foot Left decubitus)
Rows	512
Columns	512

**Table 3 animals-12-01793-t003:** Summary table of the different regression lines obtained for the two cetacean species considering the radiodensity values of the two types of bone by sections.

Portion of the Humerus	Type of Bone	N° of Section	*Stenella frontalis*	*Kogia breviceps*
Distal epiphysis of the humerus	Cortical bone	1st section	R^2^ = 0.82, *p* = 0.001	R^2^ = −0.15, *p* = 0.588
		2nd section	R^2^ = 0.81, *p* = 0.001	R^2^ = 0.24, *p* = 0.180
		3rd section	R^2^ = 0.77, *p* = 0.002	R^2^ = 0.72, *p* = 0.019
	Trabecular bone	1st section	R^2^ = 0.64, *p* = 0.001	R^2^ = 0.38, *p* = 0.113
		2nd section	R^2^ = −0.1, *p* = 0.58	R^2^ = 0.38, *p* = 0.112
		3rd section	R^2^ = −0.05, *p* = 0.45	R^2^ = 0.19, *p* = 0.211
Humerus’ body	Cortical bone	1st section	R^2^ = 0.38, *p* = 0.060	R^2^ = −0.20, *p* = 0.711
		2nd section	R^2^ = 0.16, *p* = 0.172	R^2^ = 0.16, *p* = 0.228
		3rd section	R^2^ = −0.16, *p* = 0.859	R^2^ = −0.02, *p* = 0.398
	Trabecular bone	1st section	R^2^ = −004, *p* = 0.363	R^2^ = 0.07, *p* = 0.301
		2nd section	R^2^ = −0.16, *p* = 0.864	R^2^ = 0.27, *p* = 0.16
		3rd section	R^2^ = −0.15, *p* = 0.785	R^2^ = 0.15, *p* = 0.235
Medial aspect of the humerus’ head	Cortical bone	1st section	R^2^ = 0.19, *p* = 0.154	R^2^ = 0.03, *p* = 0.340
		2nd section	R^2^ = 0.57, *p* = 0.018	R^2^ = 0.35, *p* = 0.126
		3rd section	R^2^ = 0.64, *p* = 0.010	R^2^ = 0.023, *p* = 0.349
	Trabecular bone	1st section	R^2^ = 0.19, *p* = 0.154	R^2^ = 0.24, *p* = 0.179
		2nd section	R^2^ = 0.51, *p* = 0.028	R^2^ = −0.11, *p* = 0.521
		3rd section	R^2^ = 0.70, *p* = 0.005	R^2^ = −0.18, *p* = 0.665
Lateral aspect of the humerus’ head	Cortical bone	1st section	R^2^ = 0.27, *p* = 0.103	R^2^ = 0.02, *p* = 0.348
		2nd section	R^2^ = 0.5, *p* = 0.016	R^2^ = −0.12, *p* = 0.546
		3rd section	R^2^ = 0.81, *p* = 0.001	R^2^ = 0.09, *p* = 0.285
	Trabecular bone	1st section	R^2^ = 0.42, *p* = 0.040	R^2^ = 0.17, *p* = 0.226
		2nd section	R^2^ = 0.51, *p* = 0.026	R^2^ = 0.15, *p* = 0.238
		3rd section	R^2^ = 0.5, *p* = 0.017	R^2^ = −0.14, *p* = 0.569
Proximal epiphysis of the humerus	Cortical bone	1st section	R^2^ = −0.16, *p* = 0.872	R^2^ = −0.23, *p* = 0.811
		2nd section	R^2^ = −0.19, *p* = 0.152	R^2^ = 0.09, *p* = 0.280
		3rd section	R^2^ = 0.01, *p* = 0.329	R^2^ = −0.21, *p* = 0.741
	Trabecular bone	1st section	R^2^ = −0.11, *p* = 0.602	R^2^ = −0.02, *p* = 0.401
		2nd section	R^2^ = 0.46, *p* = 0.037	R^2^ = 0.03, *p* = 0.339
		3rd section	R^2^ = −0.07, *p* = 0.513	R^2^ = 0.18, *p* = 0.215

**Table 4 animals-12-01793-t004:** Summary table of the averages and standard deviation in HU, obtained, for the different age categories of the two cetacean species, considering the radiodensity values of the two types of bone in the various sections. Abbreviations: CB, cortical bone; TB, trabecular bone.

Portion of the Bone	Type of Bone	Section	Radiodensity Average and Standard Deviation in HU				
			Atlantic Spotted Dolphin			Pigmy Sperm Whale	
			Calves	Juveniles	Adults	Calves	Adults
Distal epiphysis of the humerus	CB	1	735.1 ± 237.5	1040.3 ± 67.9	1320.9 ± 157.3	501.5 ± 62.1	516.9 ± 33.5
		2	889.7 ± 286.5	1195.5 ± 13.6	1349.7 ± 125.2	557.9 ± 78.5	572.4 ± 79.7
		3	1000.4 ± 275.7	1360.9 ± 58.8	1364.1 ± 140.5	619.5 ± 117.9	600.2 ± 55.7
	TB	1	631.7 ± 133.5	826.9 ± 87.0	903.8 ± 116.7	429.9 ± 58.3	355.0 ± 30.2
		2	662.4 ± 101.5	842.5 ± 210.1	767.5 ± 114.1	433.4 ± 52.5	372.5 ± 17.5
		3	599.6 ± 129.0	708.9 ± 98.0	706.1 ± 90.9	361.0 ± 59.1	305.3 ± 13.8
Humeral body	CB	1	1049.4 ± 238.0	1490.1 ± 118.2	1395.9 ± 143.6	717.1 ± 109.6	792.3 ± 42.1
		2	1172.1 ± 260.1	1376.4 ± 264.6	1371.9 ± 187.4	761.7 ± 101.6	899.9 ± 102.1
		3	1232.7 ± 155.8	1162.2 ± 380.4	1244.7 ± 190.3	597.3 ± 129.2	678.9 ± 4.5
	TB	1	527.1 ± 212.7	697.4 ± 16.9	657.7 ± 98.0	356.1 ± 99.2	250.0 ± 26.3
		2	499.2 ± 252.3	711.7 ± 96.1	621.6 ± 141.8	357.1 ± 48.1	263.8 ± 39.6
		3	470.4 ± 259.5	656.3 ± 84.3	586.2 ± 171.0	310.9 ± 48.3	240.6 ± 34.8
Medial aspect of the humeral head	CB	1	421.2 ± 291.1	484.3 ± 37.8	598.1 ± 149.3	283.9 ± 97.9	337.5 ± 25.9
		2	361.1 ± 246.7	519.7 ± 61.5	648.8 ± 199.4	269.4 ± 101.9	329.5 ± 30.9
		3	382.0 ± 282.5	615.6 ± 200.6	726.6 ± 234.3	181.4 ± 127.2	353.5 ± 60.8
	TB	1	336.1 ± 316.5	698.3 ± 260.7	562.7 ± 155.3	339.1 ± 34.2	303.1 ± 62.9
		2	347.3 ± 330.6	602.9 ± 119.6	601.9 ± 205.2	325.1 ± 45.2	329.2 ± 67.9
		3	470.6 ± 290.7	718.3 ± 178.1	722.2 ± 250.8	289.5 ± 30.8	367.2 ± 83.7
Lateral aspect of the humeral head	CB	1	549.5 ± 351.2	633.8 ± 63.0	719.6 ± 178.8	359.0 ± 147.5	502.4 ± 16.2
		2	592.3 ± 364.0	758.2 ± 38.1	925.1 ± 220.9	352.9 ± 151.4	480.6 ± 57.4
		3	590.2 ± 404.4	854.0 ± 225.9	1077.5 ± 184.5	246.6 ± 152.5	435.5 ± 57.8
	TB	1	401.6 ± 306.9	670.1 ± 62.7	655.6 ± 145.7	366.9 ± 35.3	336.7 ± 70.8
		2	439.8 ± 300.3	681.3 ± 129.8	700.2 ± 140.4	364.2 ± 42.9	355.1 ± 55.1
		3	560.1 ± 300.4	772.8 ± 177.7	824.5 ± 236.8	305.9 ± 55.2	386.6 ± 63.7
Proximal epiphysis of the humerus	CB	1	562.6 ± 206.9	457.4 ± 4.7	592.8 ± 159.0	243.0 ± 41.7	242.8 ± 72.5
		2	514.4 ± 99.9	416.8 ± 1.2	447.0 ± 111.8	264.2 ± 18.1	297.0 ± 75.1
		3	475.1 ± 231.2	449.1 ± 100.3	550.0 ± 173.9	380.5 ± 33.6	440.8 ± 57.0
	TB	1	384.6 ± 28.6	460.4 ± 41.8	460.6 ± 180.6	235.0 ± 40.1	191.1 ± 83.3
		2	431.0 ± 262.6	423.1 ± 53.4	404.5 ± 151.2	270.3 ± 21.1	267.7 ± 45.7
		3	475.3 ± 249.0	575.7 ± 97.4	481.8 ± 16.0	382.0 ± 27.9	343.7 ± 104.4

## Data Availability

The data presented in this study are available on request from the corresponding author.

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
