# Peer review of "Cetaceans Humerus Radiodensity by CT: A Useful Technique Differentiating between Species, Ecophysiology, and Age"

_animals, 2022, doi:10.3390/ani12141793_

Round 1

Reviewer 1 Report

The paper is very interesting from the point of view of comparative anatomy and the study of evolution. Very modern research methods have been used. The material is sufficient. The conclusions are interesting. Relevant literature. Readability of diagrams should be improved because it is poor.

Author Response

Thank you so much for your comments and suggestions.

Following the suggestion of both reviewers, the results have been rewritten in a much more readable form. Therefore, we are focusing only on the most relevant results. Furthermore, we have reduced the number of graphs and modified the table to include a summary of the statistical results as suggested.

Reviewer 2 Report

The present manuscript describes the use of bone radiodensity to compare two cetacean species' humeri and discusses potential factors for the differences found.  
Measuring radiodensity is not a novel idea, even in cetaceans. The novelty in the present manuscript resides in the potential systematicity in the method and the species used. The present work is fine in itself, but is not properly put in context with the literature, or very superficially.

General comments

The title mentions that radiodensity can be useful to differentiate species, although there are few instances where a radiodensity measure is the best way to differentiate between two species, one could argue.

This study pertains to compare bone density, therefore, other techniques such as bone histology, could have proven very interesting comparatively, to evaluate osteoblasts and osteoclasts populations in the regions singled out.

The relatively large sample size given the species is undermined by the fact that decomposition code 4 carcasses have been used. This means that in some cases only very heavily damaged carcasses were included, which implies that important information regarding the health and disease history which might have had an impact on bone density. This should be discussed more as a limitation in the discussion. This also goes for body weight absence in some cases. The results detailing the exclusion of DCC 5 are a good step in that way.

The results detailed are rather tedious to read and the HU values are of true use only in comparison. I believe the Table 3 could be implemented with an extra column with the Atlantic spotted dolphin and one other for Kogia. The results then would be more clear and only discuss the numbers instead of listing them.
Also, the proficient use of semi-colons is not necessary. Please break the sentences more with full stops.

The discussion is lacking some depth. There is no comparative mention of any literature, which would bring context.
Additional discussion on other types of uses of radiodensities in evolutionary biology such as in Gray et al., 2007 (https://anatomypubs.onlinelibrary.wiley.com/doi/epdf/10.1002/ar.20533) should be discussed in context.
Lacking also is the discussion of bone-altering diseases such as dysbaric osteonecrosis (Moore and colleagues) or brucella for example, is warranted. This aspect is thorougly disregarded.

All the figures are rather pixellated, but this could be because of the editor's compression?

Specific comments
L23: which ecophysiological aspects? this is vague
L27: ...and behavior: the Atlantic...
L28: adaptations to what? different diving depths? it is not clear at this point
L30: different individuals with different aspects? It is not clear what the authors mean here at all.
There is also no mention of the results at any point in the summary.
L39-40: this sentence is not grammatically correct, please clarify it.
L115: it is not clear whether both limbs of each animal were scanned or just one. Please clarify.
L146: the equidistant sections are not reported on Figure 1A, would it be possible to add them?
L152: the subdivision is even less clear. Please elaborate and possibly report it in Figure 1A.
Figure 1B and C should be oriented with axes.
Table 3 line 2: Humerus distal epiphysis should not be in bold.
L181: "shorter" due to age? Same question L201.
L211: "It could be observed a..." is not grammatically correct, please rephrase.
L211: "a low positive regression" is not correct. "slight positive regression" could work. Same remark on L234.
Figure 2 is far from clear. Unfortunately a lot of the data points overlap, which renders the data points difficult to separate given the same color. Maybe a color per bone section would be more suitable or shades of the same color.
L281: "with similar" slightly positive "trends"... please indicate the trends' signs. Same L285.
L305: R2 in the first section is negative. I don't understand the implication here.
L333: Specific to what? Has it previously been shown to be useful in evaluating a certain pathology or physiological process, e.g. growth, aging? Please elaborate.
L350-351: This sentence and other variations are repeated in the discussion. Maybe keep it for a conclusive sentence at the end.
L356-357: Please elaborate. The arguments are very superficially mentioned and not discussed. The same goes for the age factor.
Figure 4 would fit best in the results.
L382: notorious is incorrect. please rephrase.
L383: This should also be rephrased. It appears that the authors already knew the result. This should be presented in the introduction and materials and methods to be part of the hypothesis, proven in this case.
L386: the medial part of the humerus
L388: "_" please delete.
L395: this fact should be mentioned in table 1, and in the results (see above)
L400: it would be very interesting to see some comparative discussion here. L130 is mentioned that a review of the literature was done, but what did it bring? There is nothing on the subject in the discussion.
L401-402: this sentence lacks clarity, please rephrase.
L411: "study all species in a standard way"
L416: humeri
L416: "very different values than their counterparts" : compared to those of the other animals
L423: humeral. same for L426 and 428.
L428: in which way? increasing or reducing?
L431: ecophysiological niches.

Author Response

Response to Reviewer 2 Comments

Thank you so much for your comments and suggestions.

Please find below, point by point list of our answers.

Comments and Suggestions for Authors

  • The present manuscript describes the use of bone radiodensity to compare two cetacean species' humeri and discusses potential factors for the differences found.  
    Measuring radiodensity is not a novel idea, even in cetaceans. The novelty in the present manuscript resides in the potential systematicity in the method and the species used. The present work is fine in itself, but is not properly put in context with the literature, or very superficially.

RESPONSE:

Thank you so much for your comments and for accepting to review this manuscript. Thank you so much for your comments and for providing positive comments so our manuscript can be considered for publication in Animals. Changes were made at the end of the introduction, clarifying the reasons for undertaking this work and the project's objectives. Changes can be observed between L90 and L99.

“It is necessary to find techniques that allow evaluating how bone density is related to different diving habits and swimming styles, and to evaluate the degree of growth of an animal and therefore the bone development related to age [61]. The latest data would be important for population studies. Furthermore, it is important to create physiological reference radiodensity data for future comparative pathological studies in these species such as dysbaric osteonecrosis [62].

This work aims to validate a simple and effective technique to study the bone radiodensity of the humerus of cetaceans through CT. To approach this aim, two cetacean species (the Atlantic spotted dolphin and the pygmy sperm whale), with different swimming and diving habits, were chosen as examples.”.

General comments         

  • The title mentions that radiodensity can be useful to differentiate species, although there are few instances where a radiodensity measure is the best way to differentiate between two species, one could argue.

RESPONSE:

There are several much simpler methods to differentiate the different species starting from the simple physiognomy and external skin colouration pattern (although in cases in which only remains are found can complicate the situation). Furthermore, the article describes this technique used to differentiate the species and deepen their knowledge about them. Indeed, the technique is always described as a helpful and valid protocol, not indispensable for species differentiation.

  • This study pertains to compare bone density, therefore, other techniques such as bone histology, could have proven very interesting comparatively, to evaluate osteoblasts and osteoclasts populations in the regions singled out.

RESPONSE:

The aim of the work was not to perform a histological study. However, the samples have been taken, cuts made, and different processing and descaling protocols tested, but that study has been left, due to its volume and complexity, for another independent work.

  • The relatively large sample size given the species is undermined by the fact that decomposition code 4 carcasses have been used. This means that in some cases only very heavily damaged carcasses were included, which implies that important information regarding the health and disease history which might have had an impact on bone density. This should be discussed more as a limitation in the discussion. This also goes for body weight absence in some cases. The results detailing the exclusion of DCC 5 are a good step in that way.

RESPONSE:

We agree that the animal's health status diagnosis can be altered by the state of conservation of the carcass. Nonetheless, the work aims not to determine the cause of death but to assess the bone density of relatively healthy bones. For this reason, the study did not include animals whose differential diagnoses had conditions that could affect bone anatomy. Furthermore, all the flippers were examined with a macroscopic examination during the necropsy and with diagnostic analysis by images to discard the samples that showed signs of pathological damage. This concept has been added in the text in the materials and methods (L128-L130) and in the discussion (L371-L372).

(L128-L130)

All dissected fins were examined macroscopically during the necropsy and by CT scan. Humeri with potential bone injuries were discarded from this study”.

(L371-L372)

Those studies should also verify if these values are valid for diagnostic purposes in animals with a decomposition code of 4 or 5”.

  • The results detailed are rather tedious to read and the HU values are of true use only in comparison. I believe the Table 3 could be implemented with an extra column with the Atlantic spotted dolphin and one other for Kogia. The results then would be more clear and only discuss the numbers instead of listing them.

RESPONSE:

Following the suggestion of both reviewers, the results have been rewritten in a much more readable form. All has been focused on the regression lines considered interesting to discuss. Furthermore, the number of graphs has decreased, and we have followed your advice regarding Table 3.

  • Also, the proficient use of semi-colons is not necessary. Please break the sentences more with full stops.

RESPONSE:

Thank you so much for your comments. We will like to inform you that the punctuation and, consequently, the sentences have been simplified.

  • The discussion is lacking some depth. There is no comparative mention of any literature, which would bring context.
    Additional discussion on other types of uses of radiodensities in evolutionary biology such as in Gray et al., 2007 (https://anatomypubs.onlinelibrary.wiley.com/doi/epdf/10.1002/ar.20533) should be discussed in context.

RESPONSE:

Thank you so much for your comments and suggestion. We have proceeded to add and specify in the text that the evolution of the various species has contributed to designing particular adaptations that are more accentuated in some species than others (L311-L313). Furthermore, comparisons with what is reported in the literature have also been deepened.

(L311-L313)

“This observation agrees with the demonstration that aquatic species have undergone a readaptation of the morphology of the bone tissue with consequent weight loss during evolution [7]”.

  • Lacking also is the discussion of bone-altering diseases such as dysbaric osteonecrosis (Moore and colleagues) or brucella for example, is warranted. This aspect is thorougly disregarded.

RESPONSE:

The goal of the work was not a pathological study. Instead, as specified in the introduction, in which Moore is mentioned (L96), the work aims to validate a methodology to study the cetacean's humerus radiodensity and thus form the basis for a possible pathological study in the future. Furthermore, all the flippers were examined with a macroscopic examination during the necropsy and with diagnostic analysis by images to discard the samples that showed signs of pathological damage. This concept has been added in the text in the materials and methods (L128-L130) and in the discussion (L371-L372).

(L128-L130)

All dissected fins were examined macroscopically during the necropsy and by CT scan. Humeri with potential bone injuries were discarded from this study”.

(L371-L372)

Those studies should also verify if these values are valid for diagnostic purposes in animals with a decomposition code of 4 or 5”.

  • All the figures are rather pixellated, but this could be because of the editor's compression?

RESPONSE:

Following your recommendation, we decide to use images with the highest resolution. They are images obtained from computed tomography of the whole flipper whose sections displayed in the pictures have been enlarged.

Specific comments          

  • L23: which ecophysiological aspects? this is vague

RESPONSE:

Thank you very much for your comment. We are embracing various aspects: Species and age, which can be classified as physiological aspects; diving habits and swimming style, which can be classified as ecosystem-dependent. For this reason, we have used this term which we recognize to be vague. However, we are open to any suggestions.

  • L27: ...and behavior: the Atlantic...

RESPONSE:

Thank you very much for the comment, and we have made the necessary changes. L27

  • L28: adaptations to what? different diving depths? it is not clear at this point.

RESPONSE:

Thank you very much for the comment, and we have made the necessary changes. L33-L34

“adaptations that have developed during evolution to adapt to life in the aquatic ecosystem ”.

  • L30: different individuals with different aspects? It is not clear what the authors mean here at all.

RESPONSE:

Thank you very much for the comment, and we have made the necessary changes. L34-L36

“different characteristics such as species, age, diving behavior, and swimming style”.  

  • There is also no mention of the results at any point in the summary.

RESPONSE:

Thank you very much for the comment, and we have made the necessary changes. L28-L32

The main results encountered were significant differences between different types of bone (trabecular bone and cortical bone) and various portions of the humerus. Such differences were also obtained between species with different diving habits (shallow divers and deep divers) and distinct swimming habits (swimming style with a low-medium energy expenditure and swimming style that requires medium-high energy consumption rates)”.

  • L39-40: this sentence is not grammatically correct, please clarify it.

RESPONSE:

Thank you very much for the comment, and we have made the necessary changes.        L46-L47

Especially the cortical bone of the distal epiphysis results suggest that this bone part should be selected to be analysed in future ontogenic studies ”.

  • L115: it is not clear whether both limbs of each animal were scanned or just one. Please clarify.

RESPONSE:

Thank you very much for the comment, and we have made the necessary changes. Only one flipper for each specimen was analyzed, and I have clarified this concept in the text (L125). The reasons for this decision are budgetary/economic reasons. Furthermore, in the literature, it is reported that there is no difference between the two extremities. (de Carvalho, A. P. M. et al. Ossification Pattern of Estuarine Dolphin ( Sotalia guianensis ) Forelimbs , from the Coast of the State of Espírito Santo , Brazil. P 10, 1–10 (2015).

(L125)

Only one thoracic limb, also called a flipper”

  • L146: the equidistant sections are not reported on Figure 1A, would it be possible to add them?

RESPONSE:

Thank you very much for the comment, and we have made the necessary changes.        Figure 1A

  • L152: the subdivision is even less clear. Please elaborate and possibly report it in Figure 1A.

RESPONSE:

Thank you very much for the comment, and we have made the necessary changes. Figure 1A              

  • Figure 1B and C should be oriented with axes.

RESPONSE:        

Thank you very much for the comment, and we have made the necessary changes. (Figures 1B and 1C)

  • Table 3 line 2: Humerus distal epiphysis should not be in bold.

RESPONSE:        

Thank you very much for the comment, and we have made the necessary changes by changing the entire table 3.          

  • L181: "shorter" due to age? Same question L201.

RESPONSE:        

Thank you very much for the advice. However the results have been completely rewritten.

  • L211: "It could be observed a..." is not grammatically correct, please rephrase.

RESPONSE:

Thank you very much for the advice. However the results have been completely rewritten.

  • L211: "a low positive regression" is not correct. "slight positive regression" could work. Same remark on L234.

RESPONSE:

Thank you very much for the advice. However the results have been completely rewritten.

  • Figure 2 is far from clear. Unfortunately a lot of the data points overlap, which renders the data points difficult to separate given the same color. Maybe a color per bone section would be more suitable or shades of the same color.

RESPONSE:        

Thank you very much for the advice; we understand the problem perfectly. The problem is that there are so many variables, and the graph would be even less legible with many colours. Previous attempts have been made, and we urge the representation of the graphs as they appear. However, if the request persists, we are open and available to insert graphics with various colours.              

  • L281: "with similar" slightly positive "trends"... please indicate the trends' signs. Same L285.

RESPONSE:

Thank you very much for the advice. However the results have been completely rewritten.

  • L305: R2 in the first section is negative. I don't understand the implication here.

RESPONSE:

Thank you very much for the advice. However the results have been completely rewritten.

  • L333: Specific to what? Has it previously been shown to be useful in evaluating a certain pathology or physiological process, e.g. growth, aging? Please elaborate.

RESPONSE:

Thanks for the comment, and we have made the necessary changes. (L280-282)

Additionally, each portion was subdivided into three equidistant sections in order to evaluate if there were noticeable changes in bone radiodensity within the same portion of the bone [21,30,67,72,75]”.          

  • L350-351: This sentence and other variations are repeated in the discussion. Maybe keep it for a conclusive sentence at the end.

RESPONSE:

Thank you for the comment, and we inform you that the due changes in the structure of the discussion have been made. The cited sentence has been deleted, and the end of the discussion has been rearranged. (L366-L368)

The analyses of radiodensity on various individuals could help to understand the various characteristics that influence radiodensity, such as the individuals’ age, the species, a species-specific habits”.

  • L356-357: Please elaborate. The arguments are very superficially mentioned and not discussed. The same goes for the age factor.

RESPONSE:

Thank you very much for the comment, and we have made the necessary changes. (L304-L307)

“One of the most evident results obtained with the technique presented in this manuscript is that the humerus’ radiodensity differed between the species. The shallow-diving Atlantic spotted dolphin had a higher radiodensity than the deep-diving pygmy sperm whale.”

  • Figure 4 would fit best in the results.

RESPONSE:

Thank you very much for the comment, and we have made the necessary changes. (L258-L262)

“3.3. Comparison between species

               The results of the same sections in the two different species were compared (Figure 4). The radiodensities of the different portions of the humerus were always higher in the Atlantic spotted dolphin than in the pygmy sperm whale. The largest differences were observed at the humerus’ distal epyhisis and body (Figure 4 A and B)”.

  • L382: notorious is incorrect. please rephrase.

RESPONSE:

Thank you very much for the comment, and we have made the necessary changes. (L328)

“The differences between these two species also struck in the lateral part of the hu-merus’ head ”.

  • L383: This should also be rephrased. It appears that the authors already knew the result. This should be presented in the introduction and materials and methods to be part of the hypothesis, proven in this case.

RESPONSE:

Thank you very much for the comment, and we have made the necessary changes. (L329-L332)

“Following the study of numerous scapulohumeral joints, as a personal consideration, it is stated that the lateral humerus’ head is the part of the humerus that presents the most contact with the glenoid cavity of the scapula [2]. For this reason, this portion could be the most exposed to mechanical stress”.

  • L386: the medial part of the humerus

RESPONSE:

Thank you very much for the comment, and we have made the necessary changes. (L333)

“.. compared to the humerus’ medial part in the..”.

  • L388: "_" please delete.

RESPONSE:

Thank you very much for the comment, and we have made the necessary changes. (L335)

“…both species: highly..”.

  • L395: this fact should be mentioned in table 1, and in the results (see above)  

RESPONSE:

All humeri were measured; consequently, their longitudes can be considered results. For this reason, we did not include these values in table 1 of material and methodsHowever, now we have added a column to the table and commented on those results in the text (L190-L192).

Humerus’ length measurements showed smaller sizes in calves and juveniles age categories compared to adults. This result is because the humerus longitude increases as the animal grows from calf to adult when the size begins to stabilize”.

  • L400: it would be very interesting to see some comparative discussion here. L130 is mentioned that a review of the literature was done, but what did it bring? There is nothing on the subject in the discussion.

RESPONSE:

Thank you very much for the comment, and we have made the necessary changes. I remedied this by adding information. (L3093-L304).

“Indeed, bone density tends to increase with the animal's age and with stressful mechanisms, also in terrestrial species [14,19,26,27,30,31,69,75]”.

  • L401-402: this sentence lacks clarity, please rephrase.

RESPONSE:

Thank you very much for the comment, and we have made the necessary changes.       (L348-L349)

“In the pygmy sperm whale, the radiodensity of the cortical bone at the distal epiphysis showed a slight increase with humerus’ length and age”.

  • L411: "study all species in a standard way"

RESPONSE:

Thank you very much for the comment, and we have made the necessary changes.        (L358)

“This protocol could be used with all cetacean species…”.

  • L416: humeri

RESPONSE:

Thank you very much for the comment, and we have made the necessary changes.        (L363)

“These humeri presented very different..”.

  • L416: "very different values than their counterparts" : compared to those of the other animals

RESPONSE:

Thank you very much for the comment, and we have made the necessary changes.       (L363)

“..values compared to other animals..”.

  • L423: humeral. same for L426 and 428.

RESPONSE:

Thank you very much for the comment, and we have made the necessary changes. (L374 and L378)

“humeral radiodensity”.

  • L428: in which way? increasing or reducing?

RESPONSE:

Most of the time, an age-related increase in values ​​is described. However, as described above, there are some bone portions in which the density decrease as the animal grows. For this reason, I speak generically of age-related changes in density. (L379)

Humeral radiodensity changed with age..”.

  • L431: ecophysiological niches.

RESPONSE:

Thank you very much for the comment, and we have made the necessary changes. (L382)

“ecophysiological niches”.

Round 2

Reviewer 2 Report

The authors did make some of the required changes, such as the Table 3, which clarify the results. Most of the responses were to the point and made necessary changes. In particular the results are now much easier to read. From the previous review: • This study pertains to compare bone density, therefore, other techniques such as bone histology, could have proven very interesting comparatively, to evaluate osteoblasts and osteoclasts populations in the regions singled out. RESPONSE: The aim of the work was not to perform a histological study. However, the samples have been taken, cuts made, and different processing and descaling protocols tested, but that study has been left, due to its volume and complexity, for another independent work. Fine, then mention it in the discussion and possibly in the conclusion. My main concern remains in the superficial discusison of some topics. See below. There are still some strange aspects to the manuscript. in particular: paragraph L277-285. This reads like it should belong to the material and methods for a good part, and some in the discussion. The authors do not discuss, they state. The reader is interested in why the division in portions is worth it, what does it bring? The authors mention that some studies are based on the whole bone and that is not precise enough, but also that the studies based on one spot are also limited. So, how does this justifies their methodology? The following paragraph offers a partial answer, but why in the next paragraph? L283-285 is another statement without discussion. L336: Same general remark: highly energetic vs. lower energetic. This is not a discussion. I assume the authors imply how the flipper is involved in directional swimming and quick changes in direction that the spotted dolphins use as a hunting strategy. I am guessing it is what is meant with high energetic, but I think it could be the place to break it down, with appropriate references. L342: Same remark. The authors mention time and time again the use of bone density for age determination. What are the other ways? The authors should discuss how their technique can present advantages when other samples are not available, and other potential adavantages. Answering the question why anybody should apply it. L302-305 is an addition to the manuscript, in which the authors mention that “it is important to remember that differences in bone density due to factors such as age and stressful mechanisms have been described”. The mentioning of stressful mechanisms are not enough to just be cited twice in a row and add a string of references. Do the authors imply the notion of mechanical stress on bones? The sentence could also be rephrased. L305-309: is this related to the rest of the paragraph? More importantly, isn’t this in agreement with most of the previous literature regarding species differences in bone density? This should be mentioned here. L308: I am still not convinced by the general tone of this claim. So, like the authors mention, there are differences in bone density across species. Not new. They may be correlated to (not necessarily caused by) age, swim patterns/energy use and diving. Is this new? The main claim of this manuscript lies in the methodology. I see only one course of action here, and it is not used: Explain at least once clearly the findings of the literature review, and using your results, claim that your methodology confirms the main findings of previous works and why it is better than previous ones. This is the heart of this work. In response to the figure 2 and 3, it makes sense that different colors might not clarify the figures. I though suggest at least different shades of orange and black/gray, not so differently from Figure 4. L300-302: the authors claim that their protocol is validated by their consistent findings. This is typically represented by the variance of the results. It would be adequate then to represent this variance in Figure 2 and 3 (with bars on the datapoints for example) to present this consistency. In order for this paper to be useful to future comparative studies, I believe HU values would be useful to share. I understand this may be difficult to present though, so it is not a requirement. Specific comments L1: Cetaceans’ : no apostrophe is needed here. I would even drop the s. L24: “through computer tomography” should be at the end of the sentence. L29-30: obtained : observed L32: The obtained results: These results L33: the adaptations of this tissue to different lifestyles of the animals: “the diverging ecophysiology of” or “different ecological niches of” L39-40: I suggest something like: The consistent differences in the radiodensities of the distal epiphysis suggest that this bone portion could be particularly useful for future ontogenetic studies. L41: at this point I strongly suggest using humeral instead of humerus’ throughout the manuscript. Anatomical names are not supposed to have this apostrophe. L44: based on a review of previous studies : please insert the reference numbers here for those previous studies. L76: there is no need for “’s” in sample or animal. L79: please replace ( with [ L82: pf : of L87: animal’s : species L90: find: establish standardized L90: the evaluation of L92: “therefore” here is not logical. There is a need for techniques to evaluate the relationship between bone density and swimming style, diving habits. Why not. There is a need for techniques to evaluate bone growth. Why not. However, this does not imply that the bone development is related to age. Please clarify. L95: such as in cases of dysbaric osteonecrosis. L114: calve: calf L125: left or right? or is it maybe the same? Simply mention it. L128: what does potential mean? That the animal was positive for brucella but the bones looked normal? otherwise simply mention “animals with bone lesions…” L211:...bone, being the... : ...bone, with the… Also see L247 L233: changed: varies L265:“The graphs show the different humerus’ portion radiodensity changed in relationship with”: ...show differences in --- in relation to --- L273: The protocol presented in this study… L275: different purposes: ok, which ones? are they all discussed below? L288 Felt (1965) is L293: The brand of the software is useless here, it should remain in the materials and methods. L315: likely caused by L329: ...two species were particularly different from the lateral… L337: higher or lower joint stress, respectively. L352: biased: sampled, thus potentially introducing an age bias for this species. L362: freshness: conservation code L364: -vation code 5): code 5 (4?) and missing life and disease history). L368: question: is diet also a possible factor? Has it been considered? L372: the values are not given in the present work, except maybe in the graphs. If this has a chance at being used in healthy vs. pathological cases (although not discussed here), the data points and estimated curve should be presented properly, not only regression lines.

Author Response

Please see it in attachment.

Round 3

Reviewer 2 Report

I have no further comment.